statistics/computational biology/bioinformatics

crowd wisdom, unsupervised dimension reduction, data fusion, meta-analysis

**Author for correspondence:**
Lingfei Wang
e-mail: lingfei@broadinstitute.org

# Accurate wisdom of the crowd from unsupervised dimension reduction

## Lingfei Wang and Tom Michoel

Division of Genetics and Genomics, The Roslin Institute, The University of Edinburgh, Easter Bush, Midlothian EH25 9RG, UK

 LW, 0000-0001-9175-7006; TM, 0000-0003-4749-4725

Wisdom of the crowd, the collective intelligence from responses of multiple human or machine individuals to the same questions, can be more accurate than each individual and improve social decision-making and prediction accuracy. Crowd wisdom estimates each individual's error level and minimizes the overall error in the crowd consensus. However, with problem-specific models mostly concerning binary (yes/no) predictions, crowd wisdom remains overlooked in biomedical disciplines. Here we show, in real-world examples of transcription factor target prediction and skin cancer diagnosis, and with simulated data, that the crowd wisdom problem is analogous to one-dimensional unsupervised dimension reduction in machine learning. This provides a natural class of generalized, accurate and mature crowd wisdom solutions, such as PCA and Isomap, that can handle binary and also continuous responses, like confidence levels. They even outperform supervised-learning-based collective intelligence that is calibrated on historical performance of individuals, e.g. random forest. This study unifies crowd wisdom and unsupervised dimension reduction, and extends its applications to continuous data. As the scales of data acquisition and processing rapidly increase, especially in high-throughput sequencing and imaging, crowd wisdom can provide accurate predictions by combining multiple datasets and/or analytical methods.

## 1. Introduction and approach

The concept of wisdom of the crowd was originated in social sciences [1] well before Fisher established meta-analysis quantitatively [2]. Its philosophy has also been discovered and rediscovered in a wide range of sociological and statistical contexts, e.g. as cultural consensus [3,4] and unsupervised ensemble learning [5,6]. Recently, by extending the definition of 'individuals' of the crowd to the surging amount of datasets and analytical methods, the biomedical community is also

**Figure 1.** Example generic scenario of wisdom of the crowd predicting patient survival. (Left to right) *Patients' raw data*: different types of data are measured for a number of patients whose survival of a specific disease is to be predicted. *Individual method*: to predict patient survival, each machine or human individual has access to and/or chooses to use a subset of the measurements, based on their potentially different professional or algorithmic training. The theoretically optimal but practically unachievable method (bottom) has unlimited training data, and uses all measurements available to compute the empirical survival rate among the patients with (asymptotically) identical measurements in the training set. *Individual prediction*: each individual's prediction effectively estimates the actual survival rate, and is independently prone to errors and individual biases. *Crowd wisdom*: to obtain an accuracy higher than most or everyone of the individuals, the wisdom of the crowd combines individual predictions in the absence of raw data or information of the individual methods, by simultaneously fitting the combined prediction and the accuracy of each individual. This generic scenario also includes specific scenarios such as multiple individuals using the same raw dataset, or the same individual predicting based on different raw datasets.

seeing a wider range of applications of the crowd wisdom or meta-analysis in [7–10]. See figure 1 for an example scenario of crowd wisdom.

To estimate individual error level, crowd wisdom studies typically rely on the fundamental assumption that each individual is an independent estimator of the ground-truth, possessing their knowledge as the signal and bias as the error (figure 2a). As long as the group or ensemble of individuals remain unbiased as a whole, aggregating individual estimators for the same predictive variables would strengthen the signal and cancel out their errors. This can be regarded as a more complex version of averaging multiple measurements of the same variable.

However, previous crowd wisdom classification studies have focused predominantly on binary responses and problem-specific models [5,6,11,12], where the confusion matrices of individuals and the binary true classes were fit in turn to maximize the model's likelihood with expectation-maximization. Where available, continuous individual predictions such as confidence levels are thresholded and mostly lost, potentially limiting the classification accuracy and generalizability, while the proper choice of threshold can also be difficult. Despite some existing applications of principal component analysis (PCA) in specific research contexts [3,4,10], the general relationship between wisdom of the crowd and standard dimension reduction techniques was not understood.

To resolve these issues, we consider continuous rather than binary variables for individual responses. Due to a lack of complete information to perfectly determine the true classes, we introduce an unknown intermediate layer representing the probability of the true class given the incomplete information (*class probability* in figure 2b or, as an example, P(Survival | All data) in figure 1). In the simplest scenario, individual responses are then independent continuous estimations of the class probability. More generally, individuals can also characterize and estimate classification confidence with any other continuous scores, which are assumed to be equivalent in ranking with the class probability. Binary responses can also be treated as numerical 0s and 1s.

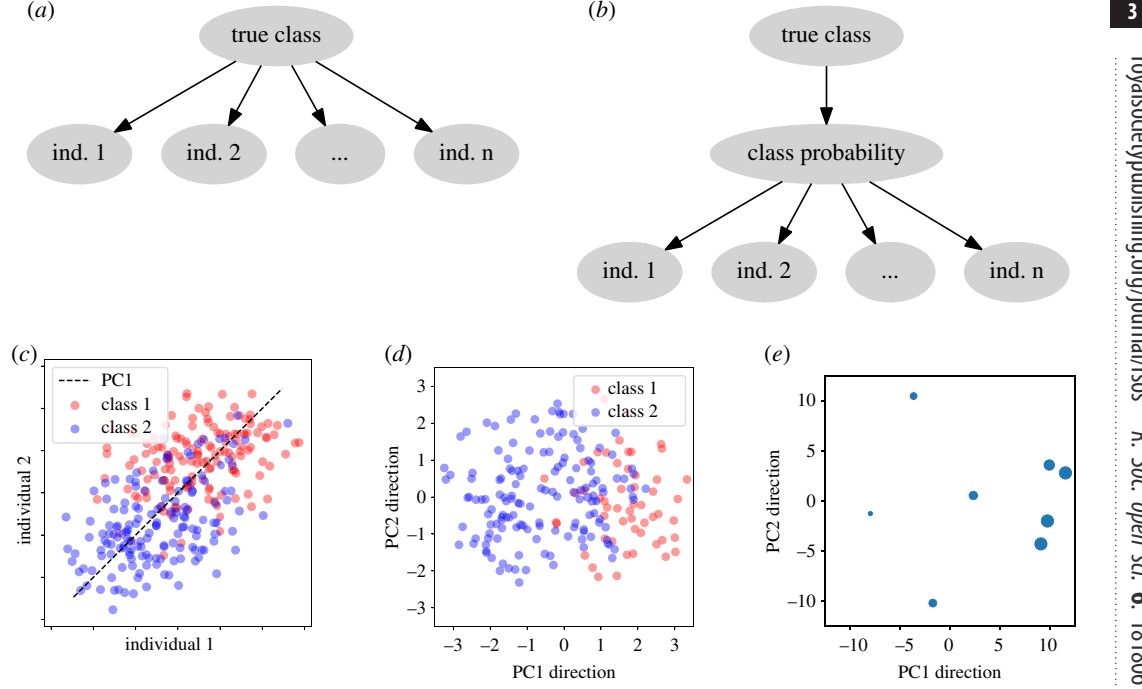

**Figure 2.** Illustrations of wisdom of the crowd. (*a*) Probablistic graph of the conventional crowd wisdom. Each individual is assumed to contain independent errors on top of the true class. (*b*) Probablistic graph of the new crowd wisdom. The intermediate continuous variable of class probability is introduced as what individuals estimate with independent errors. (*c*) Illustrative application of PCA crowd wisdom on two individuals independently estimating the class probability. (*d*,*e*) PCA recovered classification (*d*) and individual accuracy (*e*, in terms of AUROC as radius) in PC1 direction on the DREAM2 dataset.

The continuous crowd wisdom classification problem can then be solved by unsupervised dimension reduction. Unsupervised dimension reduction infers the latent lower dimensions by which the input data are assumed to be parametrized. In crowd wisdom (figure 2*b*), each individual independently estimates, and is effectively parametrized by, the class probability alone. Therefore, the class probability may be recovered as the first and only dimension (figure 2*c*, subject to a monotonic transformation). This makes dimension reduction the natural crowd wisdom for classification problems with continuous information. Which dimension reduction method is the best then depends on various aspects of the problem, such as nonlinearity. As a brief demonstration with the DREAM2 BCL6 Transcription Factor Prediction challenge dataset, containing the confidence scores of 200 genes as potential targets of BCL6 (i.e. questions) submitted by eight teams (i.e. individuals) [13–15], the first principal component (PC1) direction of the gene-by-individual matrix gave accurate representations of the class probability ranking (figure 2*d*) and the performance of each individual (figure 2*e*).

## 2. Results

To first evaluate off-the-shelf dimension reduction methods on binary responses, we envisioned an algorithm-assisted diagnostic committee of 24 dermatologists whose skin cancer classifications are known for 111 dermoscopy images [16]. As a comparison, we applied dimension reductions methods that are non-parametric (PCA, factor analysis (FA), and multi-dimensional scaling (MDS)) or nearest neighbour based (locally linear embedding (LLE), Hessian LLE, local tangent space alignment (LTSA), Isomap, and spectral embedding) to estimate the class probability ranking from the individual classifications (§3). PCA and FA were superior to most individual dermatologists and were among the top crowd wisdoms. Nearest-neighbour-based methods were not significantly more accurate than PCA, but instead converged towards PCA at large numbers of neighbours, suggesting no significant post-normalization nonlinearity (figure 3*e*). PCA and FA offered continuous confidence levels which reduced to state-of-the-art binary crowd wisdom solutions from SML [6] and CUBAM [17] at certain thresholds (figure 3*a*,*b*; electronic supplementary material, figure S1*a*,*b*). Interestingly, although a deep neural network (DNN) trained on 130 000 clinical images could outperform most dermatologists [16],

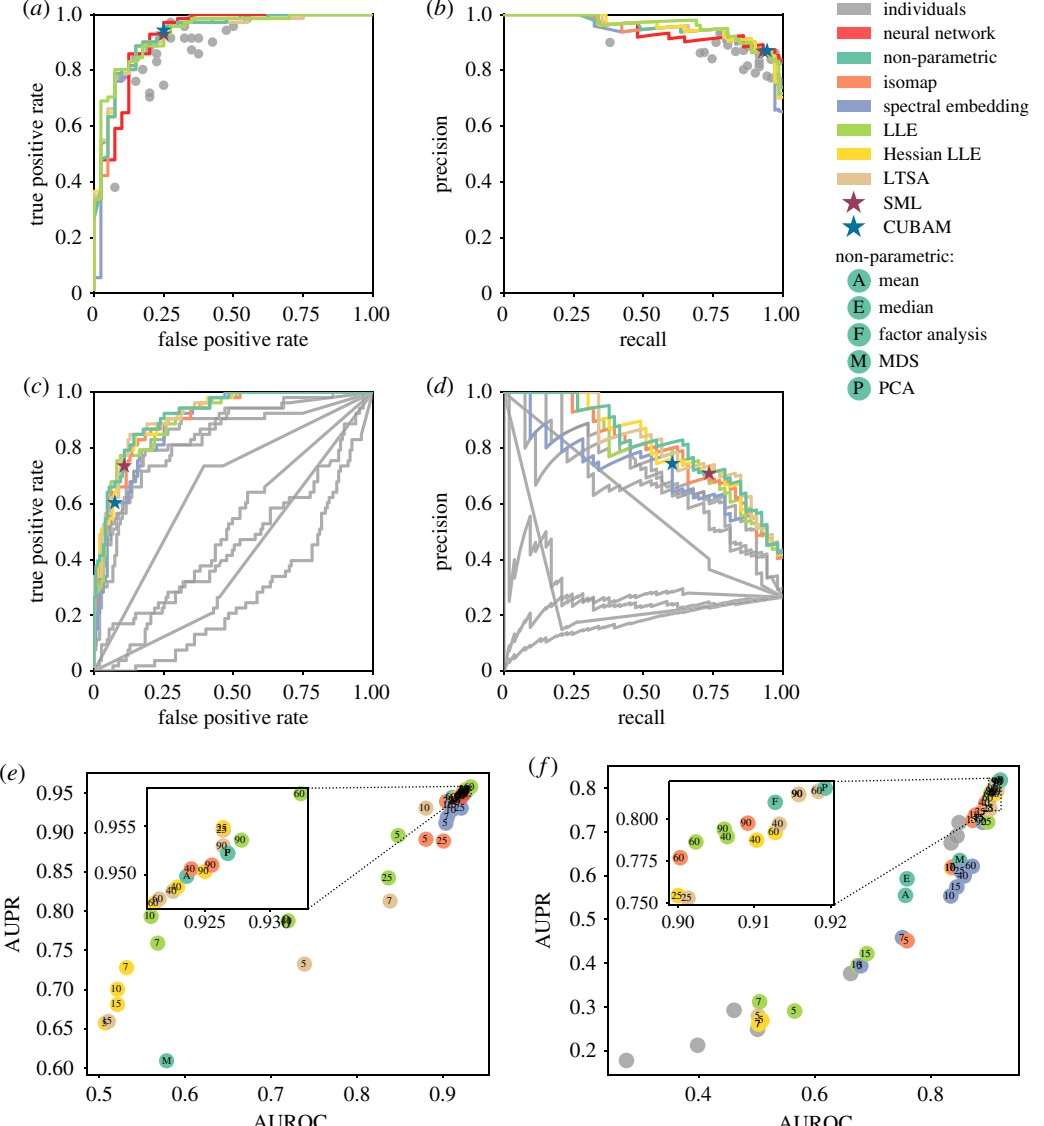

**Figure 3.** Off-the-shelf dimension reduction methods outperformed most or all individuals and existing crowd wisdoms by accounting for confidence information. (*a*–*d*) ROC (*a,c*) and Precision-Recall (*b,d*) curves and plots of individual responses, a DNN, existing crowd wisdoms, and selected dimension reduction methods for skin cancer classification (*a,b*) and the DREAM2 challenge (*c,d*). Parametric dimension reductions selected the best parameter (in *e* or *f*) according to AUROC (*a,c*) or AUPR (*b,d*). PCA is selected for non-parametric dimension reduction. SML and CUBAM only accept and output binary responses (§3). (*e,f*) AUROC and AUPR from individual responses, dimension reduction, existing crowd wisdom methods and a DNN for skin cancer classification (*e*) and the DREAM2 challenge (*f*). The inset magnifies the ranges of top 15 AUROCs and AUPRs. Numbers indicate the number of nearest neighbours.

at least 15 crowd wisdoms of those dermatologists had even better classification performance in terms of areas under the Receiver Operating Characteristic (ROC) and Precision-Recall curves (AUROC and AUPR for short, figure 3*a,b,e*; electronic supplementary material, table S1). On the other hand, treating the DNN as an extra artificial intelligence dermatologist had very minor improvement on crowd wisdom, possibly because the crowd wisdom performance was already near-optimal, close to the Bayes error rate (electronic supplementary material, figure S2 and table S1). This demonstrates the cutting-edge efficacy from off-the-shelf dimension reduction on the binary crowd wisdom task.

To test whether continuous confidence information can improve accuracy, we applied the same dimension reduction methods on the DREAM2 dataset, as well as on their perfectly binarized yes/no responses (§3). PCA on continuous confidence levels was more accurate than SML and CUBAM on binarized responses (figure 3*c,d*; electronic supplementary material, figure S3). Performance differences between crowd wisdoms were in agreement with the skin cancer classification data, except that mean

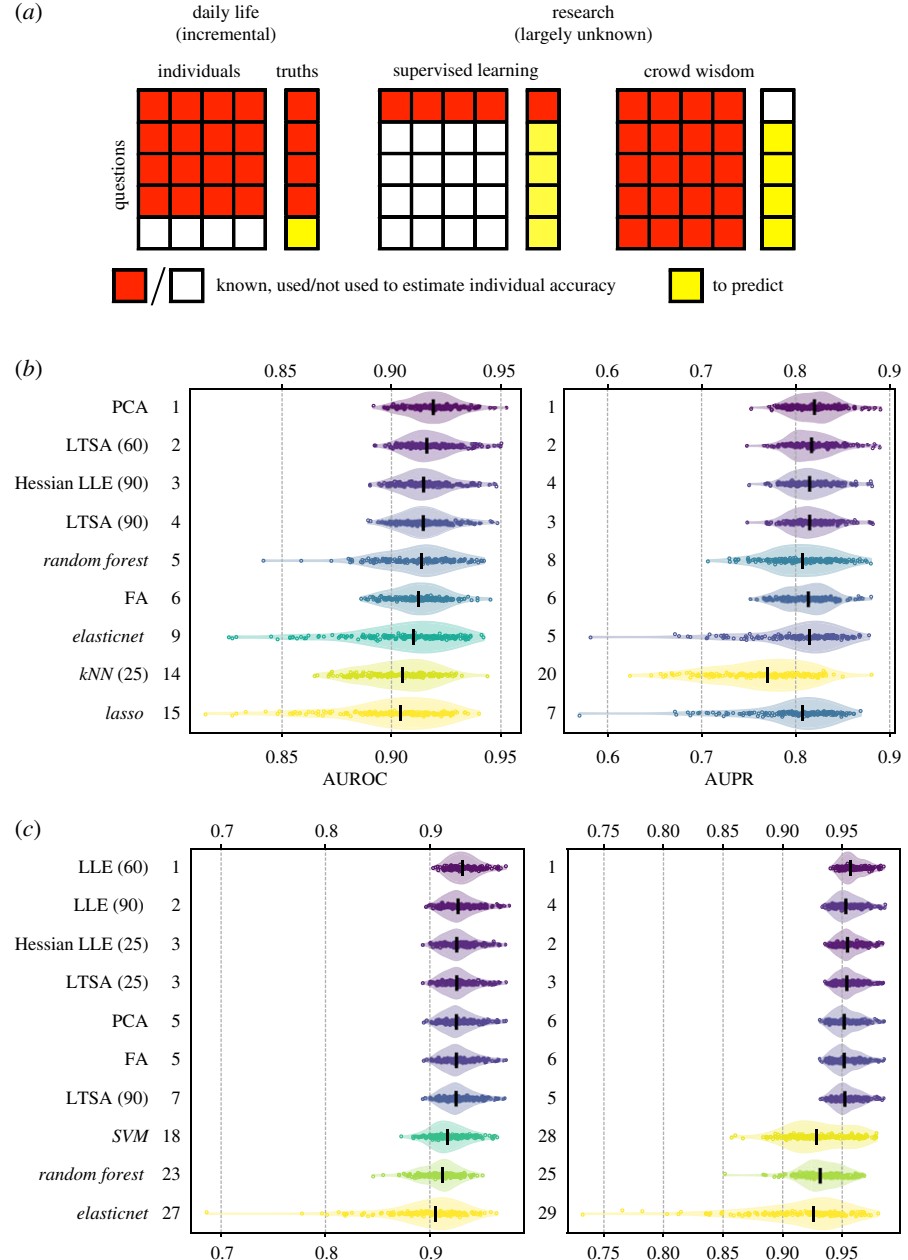

**Figure 4.** Crowd wisdom outperformed supervised learning in cross-validation. (a) Illustration of the difference between research (right) and daily life (left), in that ground-truths are largely unknown. Therefore, crowd wisdom may incorporate more information and estimate individual accuracies better than supervised classifiers. (b,c) Empirical distributions and medians of AUROC (left) and AUPR (right) of the top 5 crowd wisdom and three supervised learning methods (in either AUROC and AUPR) in 200 cross-validations with 25% random partition of training data are shown for the DREAM2 (b) and skin cancer (c) datasets. Method names include the numbers of nearest neighbours in brackets, and are *italicized* for supervised classifiers. Numbers next to the frames represent rankings of the methods in terms of median AUROC or AUPR among all 58(b)/56(c) methods, which are also reflected in colouring.

and median—often the default crowd wisdom method for continuous data [7]—could not account for worse-than-random individuals (figure 3c,d,f; electronic supplementary material, figures S1c,d and S4). Many dimension reduction methods, such as PCA and Isomap, outperformed every team. Dimension reduction provided reliable and superior crowd wisdom by accounting for confidence information.

Knowledge of the ground-truth for a subset of questions may help calibrating response aggregations for the remaining questions. For instance, in daily life we trust people and favour programmes that were more accurate historically, although the scenario is remarkably different in research (figure 4a).

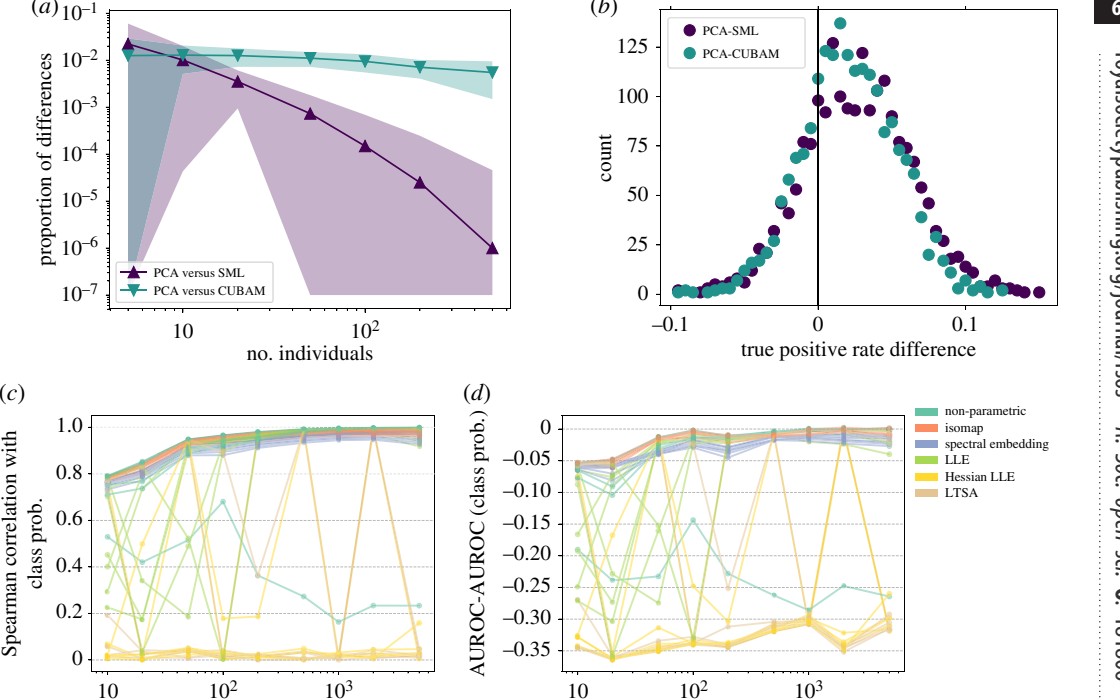

**Figure 5.** Dimension reduction methods outperformed existing binary crowd wisdom methods and recovered the class probability. (a) The proportion of differences in aggregated responses as a function of the number of individuals on simulated binary datasets. Markers and shades represent mean and standard deviation in 2000 random repeats of 1000 questions each. (b) Empirical distribution of true positive rate differences between PCA and existing binary crowd wisdoms at their respective false positive rate thresholds in 2000 random repeats of simulation 1. (c,d) Absolute Spearman correlation (c) and AUROC difference (d) between each crowd wisdom and the true class probability at different numbers of individuals. See §3 for details.

To compare calibrated response aggregations against ground-truth-ignorant crowd wisdoms, we performed 200 cross-validations of the same crowd wisdoms and eight popular supervised classifiers (including linear, logistic, lasso and elasticnet regression, linear discriminant analysis (LDA), support vector machine (SVM), kNN and random forest) that were trained on randomly selected question subsets (10–90%, §3). As expected, at low training set proportions (typically less than 50%), crowd wisdom had better performance in terms of AUROC and AUPR than supervised classifiers for both the DREAM2 and the skin cancer datasets (figure 4b,c; electronic supplementary material, figures S5 and S6). Supervised classifiers could only compare with crowd wisdom when the training dataset was bigger than the test set (electronic supplementary material, figures S5 and S6). Considering that the true answers in practical research questions are mostly unknown, unsupervised crowd wisdom outperformed supervised learning by integrating the test dataset to better estimate individual accuracies.

We further interrogated crowd wisdoms in controlled simulations. With 2000 replicated simulations for each parameter set, we found SML to highly correlate with and converge to thresholded PCA on binary/binarized datasets as the number of individuals increased (figure 5a, §3). With 500 individuals, only 2 out of 2 000 000 predictions were different (figure 5a), possibly due to SML's similarity with PCA by design [6]. Consequently, due to loss of information in the (even perfect) binarization of confidence levels, SML was less sensitive than PCA on continuous data (figure 5b, Student's $t$-test $p < 10^{-189}$ and Wilcoxon test $p < 10^{-162}$; electronic supplementary material, figure S7, §3). CUBAM did not converge to PCA (figure 5a; electronic supplementary material, figure S8) due to its nonlinearity, and it was also less sensitive than PCA after binarization (figure 5b, Student's $t$-test $p < 10^{-161}$ and Wilcoxon test $p < 10^{-142}$; electronic supplementary material, figure S7).

In individual simulations (§3), PCA, FA, Isomap and LLE converged to perfect class probability predictions as the number of individuals increased (figure 5c,d; electronic supplementary material, figure S9b–i), but LLE-based methods were unreliable on noisy datasets (electronic supplementary material, figure S9j,k, [18]). Individual simulations also reaffirmed our existing conclusions. PCA, FA and Isomap continued to lead the performances (electronic supplementary material, figures S10 and S9) and crowd wisdom remained superior to supervised classifiers (electronic supplementary material,

figures S10*b* and S11). Mean and median were again hindered by worse-than-random individuals (electronic supplementary material, figures S10*a* and S9). Overall, PCA and Isomap were more reliable and accurate than other dimension reduction methods and previous wisdom of the crowd methods.

# 3. Methods

## 3.1. DREAM2 BCL6 Transcription Factor Prediction Challenge dataset

The DREAM2 BCL6 Transcription Factor Prediction Challenge is an open crowd challenge to infer BCL6 gene's transcriptional targets [13–15]. Participating teams inferred BCL6 targets from gene expression microarray and optional external data, and submitted confidence scores for 200 potential target genes. Submissions were evaluated against the gold standard derived from ChIP-on-chip and perturbation experiments, containing 53 BCL6 targets. We had access to submissions from 11 teams, in which eight were full (without missing predictions) and were used for crowd wisdom.

## 3.2. Skin cancer classification dataset

DNNs outperformed an average dermatologist in the classification of skin cancer from dermoscopy images [16]. Based on dermoscopy images alone, dermatologists were asked whether to biopsy/treat the lesion or to reassure the patient. We obtained 24 dermatologists' responses to 111 biopsy-proven dermoscopy images in which 71 were malignant. We also obtained the predicted confidence scores for these images from the DNN in [16].

## 3.3. Simulated datasets

A simulated dataset of $n$ binary (yes/no) questions contains their true classes, the (posterior) class probabilities given all the relevant data for each question as $P_i(Yes \mid data)$, and the responses from $k$ individuals to all $n$ questions as matrix $\mathbf{R} = \{r_{ji}\}$, for $i = 1, \ldots, n, j = 1, \ldots, k$. Given the desired occurrence frequency of class yes as $P(Yes)$, the dataset needs to contain $nP(Yes)$ questions in class yes and $n(1 - P(Yes))$ in class no. We simulated the true classes, class probabilities, and individual responses (figure 2*b*) according to the following steps:

(1)  Simulate class probabilities $P(Yes \mid data) \sim B(\beta, \beta)$, where $B$ is the Beta distribution and $\beta$ characterizes the question difficulty given all the data. For each question, set the true class to yes with probability $P(Yes \mid data)$ and no otherwise. Only the first $nP(Yes)$ questions in yes class and the first $n(1 - P(Yes))$ questions in no class were retained, merged, and shuffled to form the full list of questions $i = 1, \ldots, n$. Their class probability $P_i(Yes \mid data)$ and true classes were recorded.
(2)  Simulate individual responses $\mathbf{R}$. Individual $j$'s response to question $i$ is $r_{ji} \sim N(\alpha_j P_i(Yes \mid data), 1)$, where $\alpha_j \sim N(\bar{\alpha}, \sigma_\alpha^2)$.
(3)  Normalization was applied (cf below).

The simulation takes six parameters: $k, n, P(Yes), \beta, \bar{\alpha}$ and $\sigma_\alpha$. See electronic supplementary material, table S2 for parameter values.

## 3.4. Perfect binarization

To transform confidence level responses to binary (yes/no) responses, we chose the ideal scenario for existing binary crowd wisdom methods, by assuming that each individual knows the true total number of yes responses. Consequently, each individual will select that same number of their most confident predictions as yes, and the rest as no. Ties at the yes/no boundary are selected at random.

## 3.5. Normalization

We normalized raw answers from multiple individuals to multiple questions before applying crowd wisdom or supervised learning (in cross validation). For continuous datasets, we first converted raw answers into rankings, separately for each individual and with ties averaged. Then, for all datasets, we shifted the raw or rank-converted values to zero mean and scaled them to unit variance, separately for each individual.

## 3.6. Dimension reduction as wisdom of the crowd

From the python package *scikit-learn*, we applied the following dimension reduction methods for crowd wisdom: *TruncatedSVD* (as a part of PCA) and *FactorAnalysis* in *sklearn.decomposition*, and *LocallyLinearEmbedding* (with methods standard, hessian and ltsa), *Isomap* and *SpectralEmbedding* in *sklearn.manifold*. Nearest-neighbour based methods took 5, 7, 10, 15, 25, 40, 60 and 90 neighbours. We also included mean and median as simple statistics for crowd wisdom.

## 3.7. Evaluation metrics

We used the Receiver Operating Characteristic and Precision-Recall curves, as well as their areas under the curves (AUROC and AUPR) as evaluation metrics. To tackle the sign indeterminacy from dimension reduction, we always computed these metrics twice, on the original output and on its negative, and selected the one with a larger area under the curve for comparison. For fair comparison, the same procedure was applied on supervised learning methods. In practice, sign indeterminancy can be solved by assuming more than half of the individuals have better-than-random responses, and then aligning crowd wisdom with the majority of the crowd.

## 3.8. Supervised classifiers

From the python package *scikit-learn*, we applied the following supervised classifiers: *LinearRegression*, *ElasticNetCV*, *LassoCV* and *LogisticRegression* in *sklearn.linear_model*, *LinearDiscriminantAnalysis* in *sklearn.discriminant_analysis*, *RandomForestClassifier* in *sklearn.ensemble* and *KNeighborsClassifier* in *sklearn.neighbors* with 5, 7, 10, 15, 25, 40, 60 and 90 neighbours.

## 3.9. Method comparison in cross validation

To compare crowd wisdom and supervised classifiers, we randomly split each dataset into a training set (containing 10, 20, 25, 40, 60, 80 or 90% of all questions) and a test set (for the rest), using *sklearn.model_selection.StratifiedShuffleSplit* and requiring the number of questions to be larger than that of individuals in the training set. Supervised classifiers were trained on individual predictions against ground-truths in the training set, and then predicted for the test set. For crowd wisdom, we performed crowd wisdom on the full data (not using ground-truth) and then extracted predictions for the test set. Evaluation metrics were computed for every random split. The random split was repeated 200 times per split ratio per dataset.

## 3.10. Method comparison on binarized data

With a given parameter set for simulation, we performed 2000 replicated simulations with different random seeds. For each replicate, the ROC curve for PCA and the FPR and TPR for SML and CUBAM were computed. The ROC quantiles were computed as the quantiles of TPR at every FPR level among the 2000 ROCs from replicates. The densities of SML and CUBAM points on ROC were computed with *scipy.stats.gaussian_kde*. The TPR difference between PCA and SML or CUBAM was computed at SML's or CUBAM's FPR in each replicate.

## 3.11. Proportion of differences between binary crowd wisdoms and PCA

Simulations 21, 22, 23, 19, 1, 20, 25 and 24, 3, 1, 4, 5, 6, 7 were used, respectively, for the comparisons of differences as the numbers of questions and individuals vary. Each simulation consists of 2000 replications with different random seeds. Within each replication, the binary crowd wisdom (SML or CUBAM) and PCA were first applied on the binary(/binarized) simulated data. Should the AUROC between PC1 of PCA and the binary crowd wisdom be less than 0.5, the signs of PC1 are inverted. PC1 is then thresholded so the largest $N$ entries are positive, in which $N$ is the number of positives from the binary crowd wisdom. The proportion of differences is the number of questions on which the thresholded PC1 and the binary crowd wisdom have different predictions, divided by the total number of questions. The mean and standard deviation were then computed across the 2000 replicates, after excluding the (rare) critical failures or single-valued outputs of the binary crowd wisdom.

# 4. Discussion

We embed the question of wisdom of the crowd in unsupervised dimension reduction, and allow them to co-develop. This makes crowd wisdom classification possible on continuous datasets, with a class of generic algorithms in which PCA and Isomap are found efficient, accurate and consistent. Unsupervised dimension reductions also obtain superior performances over calibrated crowd wisdoms from supervised classifiers.

This study does not consider datasets with missing values [5], strongly correlated errors between individuals [19–22], extra information [23] or post-crowd-wisdom thresholding. Existing crowd wisdom methods, e.g. SML and CUBAM, may also have the potential to be repurposed for, or be regarded as a form of, dimension reduction. Future research on these problems within the dimension reduction framework may further widen the applications of crowd wisdom.

Data accessibility. The DREAM2 challenge dataset is available at [14]. The skin cancer classification dataset is published in [16]. Softwares for unsupervised dimension reduction are available in the python package *sklearn*.

Authors' contributions. L.W. formulated the question, designed, performed and wrote the paper. T.M. designed and supervised the study, wrote the paper and acquired funding. Both authors gave final approval for publication.

Competing interests. We declare we have no competing interests.

Funding. This work is supported by BBSRC (grant nos. BB/P013732/1 and BB/M020053/1).

Acknowledgements. L.W. thank Gustavo Stolovitzky and Robert Vogel for providing the DREAM2 dataset, and Andre Esteva and Sebastian Thrun for providing the skin cancer dataset.

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
