## [Reviewer comments · Royal Society Open Science]

Review History

RSOS-181806.R0 (Original submission)

Review form: Reviewer 1 (Simon Wilshin)

Is the manuscript scientifically sound in its present form?

Yes

Are the interpretations and conclusions justified by the results?

Yes

Is the language acceptable?

Yes

Is it clear how to access all supporting data?

No

Do you have any ethical concerns with this paper?

No

Have you any concerns about statistical analyses in this paper?

No

Recommendation?

Accept with minor revision (please list in comments)

Comments to the Author(s)

Summary

In this paper the authors demonstrate that unsupervised dimensionality reduction techniques can be used to perform wisdom of the crowds style aggregation with performance comparable or superior to that of established techniques. This aggregation can be performed with both categorical and continuous responses, with PCA and Isomap performing well. The paper is generally clear, and the figures illustrative. I have only minor comments I would like to see addressed.

Minor Comments

Pg 5 ln 5 "at least 15 crowd wisdoms of the same dermatologists" - This seems like an odd phrase. I assume this means a pooled population of at least 15.

Pg 10 ln 45 I would be useful to have a brief definition of AUROC and AUPR on pg 5 when they are first mentioned, pg 10 is a little late.

Figures

In general the ROC and PR curves are hard to distinguish and have lots of white-space. It might be beneficial to include insets for some figures where differences between curves are highlighted in the text, a small alpha on the curves might also be considered to aid the eye in following them. It could also be made clearer what the SML and CUBAN markers in Figure 3 refer to, the text suggests that their performance in each case ends up similar to another technique, but what is being labelled by the stars in the figure is not obvious.

Review form: Reviewer 2**Is the manuscript scientifically sound in its present form?**

No

Are the interpretations and conclusions justified by the results?

Yes

Is the language acceptable?

Yes

Is it clear how to access all supporting data?

Yes

Do you have any ethical concerns with this paper?

No

Have you any concerns about statistical analyses in this paper?

No

Recommendation?

Reject

Comments to the Author(s)

This manuscript describes an unsupervised dimensionality reduction approach to wisdom-of-crowd, meta-classifier learning. The authors propose using the first principal component of the columns of the prediction matrix (classifier-by-data instance) as the consensus prediction. The authors compare their PCA-based approach, along with other standard dimension reduction methods, to supervised techniques as well as two existing methods specialized for this task, SML and CUBAM, on DREAM2 transcription factor activity prediction data and skin cancer classification data. The authors conclude PCA consistently achieves best performance and has the ability to handle continuous confidence scores as input. The manuscript, though succinct, contains extensive computational experiments and is overall clearly written.

Unfortunately, the PCA-based approach proposed by the authors is near identical to the existing work of SML, which puts the novelty of this work under question. Note that SML forms the covariance matrix of the prediction matrix, replaces the diagonal entries, and finds the leading eigenvector. This eigenvector is then multiplied to the prediction matrix to obtain the consensus predictions. If one removes from SML the modification of diagonal elements of covariance matrix, the resulting algorithm is identical to the authors' PCA approach. None of this connection discussed in the manuscript, which seems misleading.

Related to this concern, the main motivation that the authors give for using dimension reduction is not sound. The authors assume a probabilistic model of the noisy ensemble of predictions where there is a single latent component that each classifier independently estimates. However, no justification is given as to how particular dimension reduction techniques that the authors consider correctly infers this latent factor, which would depend on the details of the error model and the assumptions of different dimension reduction methods. For example, PCA finds a direction that maximizes variance explained--does this objective coincide with the inference of the latent factor in the authors' model? In this regard, SML appears to be a more principled counterpart to the authors' PCA approach as SML defines a probabilistic model, and relates the leading eigenvector of the modified covariance matrix to the desired latent variable. In fact, the modification of diagonal elements, which may be the only difference between SML and PCA, is theoretically justified.

Given this observation, I was not surprised to see that SML lies exactly on PCA's performance curve in Figure 3. Indeed, the authors note that the performance of SML and PCA converge as more classifiers are added to the data set, which is likely due to the diminishing contribution of the diagonal elements on the covariance matrix on the leading eigenvector.

The authors describe the ability to handle continuous input as a major contribution, but it was not clear to me why the same SML procedure could not be applied to the continuous prediction matrix to be compared with PCA, even if SML is originally designed for binary classification. This comparison should be included in future experiments. In addition, Figure 3 should include the complete ROC or PR curves for SML using the continuous predictions scores before binarization.

Decision letter (RSOS-181806.R0)

18-Feb-2019

Dear Mr Wang,

The editors assigned to your paper ("Accurate wisdom of the crowd from unsupervised dimension reduction") have now received comments from reviewers. We would like you to revise your paper in accordance with the referee and Associate Editor suggestions which can be found below (not including confidential reports to the Editor). Please note this decision does not guarantee eventual acceptance.

Note: while Royal Society Open Science does not make judgments on 'novelty' per se, the Editors do require that a paper represents a meaningful contribution to the existing body of literature.

Please submit a copy of your revised paper before 13-Mar-2019. Please note that the revision deadline will expire at 00.00am on this date. If we do not hear from you within this time then it will be assumed that the paper has been withdrawn. In exceptional circumstances, extensions may be possible if agreed with the Editorial Office in advance. We do not allow multiple rounds of revision so we urge you to make every effort to fully address all of the comments at this stage. If deemed necessary by the Editors, your manuscript will be sent back to one or more of the original reviewers for assessment. If the original reviewers are not available, we may invite new reviewers.

- Data accessibility

It is a condition of publication that all supporting data are made available either as supplementary information or preferably in a suitable permanent repository. The data accessibility section should state where the article's supporting data can be accessed. This section should also include details, where possible of where to access other relevant research materials such as statistical tools, protocols, software etc can be accessed. If the data have been deposited in an external repository this section should list the database, accession number and link to the DOI for all data from the article that have been made publicly available. Data sets that have been

deposited in an external repository and have a DOI should also be appropriately cited in the manuscript and included in the reference list.

If you wish to submit your supporting data or code to Dryad (<http://datadryad.org/>), or modify your current submission to dryad, please use the following link:
<http://datadryad.org/submit?journalID=RSOS&manu=RSOS-181806>

- **Competing interests**

- **Authors' contributions**

- **Acknowledgements**

- **Funding statement**

on behalf of Dr Hamed Haddadi (Associate Editor) and Marta Kwiatkowska (Subject Editor)
openscience@royalsociety.org

Associate Editor's comments (Dr Hamed Haddadi):

There are concerns about the novelty of the work and presentation of the results, please see the reviews and revise the manuscript accordingly.

Comments to Author:

Reviewers' Comments to Author:

Reviewer: 1

Comments to the Author(s)

Summary

In this paper the authors demonstrate that unsupervised dimensionality reduction techniques can be used to perform wisdom of the crowds style aggregation with performance comparable or superior to that of established techniques. This aggregation can be performed with both categorical and continuous responses, with PCA and Isomap performing well. The paper is generally clear, and the figures illustrative. I have only minor comments I would like to see addressed.

Minor Comments

Pg 5 ln 5 "at least 15 crowd wisdoms of the same dermatologists" - This seems like an odd phrase. I assume this means a pooled population of at least 15.

Pg 10 ln 45 I would be useful to have a brief definition of AUROC and AUPR on pg 5 when they are first mentioned, pg 10 is a little late.

Figures

In general the ROC and PR curves are hard to distinguish and have lots of white-space. It might be beneficial to include insets for some figures where differences between curves are highlighted in the text, a small alpha on the curves might also be considered to aid the eye in following them. It could also be made clearer what the SML and CUBAN markers in Figure 3 refer to, the text suggests that their performance in each case ends up similar to another technique, but what is being labelled by the stars in the figure is not obvious.

Reviewer: 2

Comments to the Author(s)

This manuscript describes an unsupervised dimensionality reduction approach to wisdom-of-crowd, meta-classifier learning. The authors propose using the first principal component of the columns of the prediction matrix (classifier-by-data instance) as the consensus prediction. The authors compare their PCA-based approach, along with other standard dimension reduction methods, to supervised techniques as well as two existing methods specialized for this task, SML and CUBAM, on DREAM2 transcription factor activity prediction data and skin cancer classification data. The authors conclude PCA consistently achieves best performance and has the ability to handle continuous confidence scores as input. The manuscript, though succinct, contains extensive computational experiments and is overall clearly written.

Unfortunately, the PCA-based approach proposed by the authors is near identical to the existing work of SML, which puts the novelty of this work under question. Note that SML forms the covariance matrix of the prediction matrix, replaces the diagonal entries, and finds the leading eigenvector. This eigenvector is then multiplied to the prediction matrix to obtain the consensus predictions. If one removes from SML the modification of diagonal elements of covariance matrix, the resulting algorithm is identical to the authors' PCA approach. None of this connection discussed in the manuscript, which seems misleading.

Related to this concern, the main motivation that the authors give for using dimension reduction is not sound. The authors assume a probabilistic model of the noisy ensemble of predictions where there is a single latent component that each classifier independently estimates. However, no justification is given as to how particular dimension reduction techniques that the authors consider correctly infers this latent factor, which would depend on the details of the error model and the assumptions of different dimension reduction methods. For example, PCA finds a direction that maximizes variance explained--does this objective coincide with the inference of the latent factor in the authors' model? In this regard, SML appears to be a more principled counterpart to the authors' PCA approach as SML defines a probabilistic model, and relates the leading eigenvector of the modified covariance matrix to the desired latent variable. In fact, the modification of diagonal elements, which may be the only difference between SML and PCA, is theoretically justified.

Given this observation, I was not surprised to see that SML lies exactly on PCA's performance curve in Figure 3. Indeed, the authors note that the performance of SML and PCA converge as more classifiers are added to the data set, which is likely due to the diminishing contribution of the diagonal elements on the covariance matrix on the leading eigenvector.

The authors describe the ability to handle continuous input as a major contribution, but it was not clear to me why the same SML procedure could not be applied to the continuous prediction matrix to be compared with PCA, even if SML is originally designed for binary classification. This comparison should be included in future experiments. In addition, Figure 3 should include the complete ROC or PR curves for SML using the continuous predictions scores before binarization.

Author's Response to Decision Letter for (RSOS-181806.R0)

See Appendix A.

RSOS-181806.R1 (Revision)

Review form: Reviewer 2

Is the manuscript scientifically sound in its present form?

Yes

Are the interpretations and conclusions justified by the results?

No

Is the language acceptable?

Yes

Is it clear how to access all supporting data?

No

Do you have any ethical concerns with this paper?

No

Have you any concerns about statistical analyses in this paper?

No

Recommendation?

Major revision is needed (please make suggestions in comments)

Comments to the Author(s)

I agree with the authors that casting crowd-of-wisdom as dimensionality reduction and using off-the-shelf techniques like PCA is a valuable perspective that should see the light of day. However, the way the existing work of SML is presented is misleading for the readers, given that SML is grounded in spectral theory and closely resembles PCA, thus fitting into the same dimensionality reduction framework that the authors describe as novel. In response to this comment the authors have added only a high-level mention of "SML's similarity with PCA by design," but this similarity is not explained anywhere in the manuscript. In addition to adding a more detailed discussion of this theoretical connection, the language should be changed throughout the manuscript to place more emphasis on "off-the-shelf" or "standard" dimensionality reduction techniques, which I believe is the main focus of this work. I believe this update will help to emphasize the contribution of the authors.

In response to my other comment that the point estimates for SML and CUBAM in Figure 3 should be replaced with full curves for more comprehensive comparisons, the authors opted not to make this change and have added an explanation that these methods "only accept and output binary responses." Although this statement may be pedantically true, all it takes is a cosmetic change to evaluate SML and CUBAM based on their intermediate non-binary scores just before the final binarization of the output, and I have difficulty understanding why the authors are unable to incorporate this change. My initial review pointed out this possibility for SML only, but I have additionally observed that CUBAM also internally calculates non-binary scores before binarizing them, which can be similarly used to plot the entire PR/ROC curves for more informative comparison with other methods, which I believe is in the interest of the readers.

Minor comments:

- "Despite some existing applications of principal component analysis (PCA) in specific research contexts ([3, 4, 10]), the general relation between wisdom of the crowd and machine learning was not understood." -- This statement seems misleading due to its scope. Instead of "machine learning", I suggest "standard dimensionality reduction techniques."

- "We found SML to highly correlate with and converge to thresholded PCA on binary/binarized datasets as the number of individuals increased (Figure 5A, Figure S7, Methods)" -- Figure S7 states they did *not* converge. The description needs to be updated.

- The last sentence of Results section -- "Overall, PCA and Isomap were more reliable and accurate than other dimension reduction methods and previous wisdom of the crowd methods." -- does not follow logically from previous text. The same paragraph only describes comparison with supervised/other off-the-shelf methods but not previous wisdom-of-crowd methods. The previous paragraph in fact states that SML/CUBAM were less sensitive (which, as noted, is unsurprising given binarization).

Decision letter (RSOS-181806.R1)

08-Apr-2019

Dear Mr Wang:

Manuscript ID RSOS-181806.R1 entitled "Accurate wisdom of the crowd from unsupervised dimension reduction" which you submitted to Royal Society Open Science, has been reviewed. The comments of the reviewer(s) are included at the bottom of this letter.

Please submit a copy of your revised paper before 01-May-2019. Please note that the revision deadline will expire at 00.00am on this date. If we do not hear from you within this time then it will be assumed that the paper has been withdrawn. In exceptional circumstances, extensions may be possible if agreed with the Editorial Office in advance. We do not allow multiple rounds of revision so we urge you to make every effort to fully address all of the comments at this stage. If deemed necessary by the Editors, your manuscript will be sent back to one or more of the original reviewers for assessment. If the original reviewers are not available we may invite new reviewers.

- Ethics statement

- Data accessibility

- Competing interests

- Authors' contributions

- Acknowledgements

- Funding statement

on behalf of Dr Hamed Haddadi (Associate Editor) and Professor Marta Kwiatkowska (Subject Editor)
openscience@royalsociety.org

Associate Editor Comments to Author (Dr Hamed Haddadi):

Please observe the reviewer's comment and address for an updated submission.

Reviewer comments to Author:

Reviewer: 2

Comments to the Author(s)

I agree with the authors that casting crowd-of-wisdom as dimensionality reduction and using off-the-shelf techniques like PCA is a valuable perspective that should see the light of day. However, the way the existing work of SML is presented is misleading for the readers, given that SML is grounded in spectral theory and closely resembles PCA, thus fitting into the same dimensionality reduction framework that the authors describe as novel. In response to this comment the authors have added only a high-level mention of "SML's similarity with PCA by design," but this

similarity is not explained anywhere in the manuscript. In addition to adding a more detailed discussion of this theoretical connection, the language should be changed throughout the manuscript to place more emphasis on “off-the-shelf” or “standard” dimensionality reduction techniques, which I believe is the main focus of this work. I believe this update will help to emphasize the contribution of the authors.

In response to my other comment that the point estimates for SML and CUBAM in Figure 3 should be replaced with full curves for more comprehensive comparisons, the authors opted not to make this change and have added an explanation that these methods “only accept and output binary responses.” Although this statement may be pedantically true, all it takes is a cosmetic change to evaluate SML and CUBAM based on their intermediate non-binary scores just before the final binarization of the output, and I have difficulty understanding why the authors are unable to incorporate this change. My initial review pointed out this possibility for SML only, but I have additionally observed that CUBAM also internally calculates non-binary scores before binarizing them, which can be similarly used to plot the entire PR/ROC curves for more informative comparison with other methods, which I believe is in the interest of the readers.

Minor comments:

- “Despite some existing applications of principal component analysis (PCA) in specific research contexts ([3, 4, 10]), the general relation between wisdom of the crowd and machine learning was not understood.” -- This statement seems misleading due to its scope. Instead of “machine learning”, I suggest “standard dimensionality reduction techniques.”

- “We found SML to highly correlate with and converge to thresholded PCA on binary/binarized datasets as the number of individuals increased (Figure 5A, Figure S7, Methods)” -- Figure S7 states they did *not* converge. The description needs to be updated.

- The last sentence of Results section -- “Overall, PCA and Isomap were more reliable and accurate than other dimension reduction methods and previous wisdom of the crowd methods.” -- does not follow logically from previous text. The same paragraph only describes comparison with supervised/other off-the-shelf methods but not previous wisdom-of-crowd methods. The previous paragraph in fact states that SML/CUBAM were less sensitive (which, as noted, is unsurprising given binarization).

Author's Response to Decision Letter for (RSOS-181806.R1)

See Appendix B.

Decision letter (RSOS-181806.R2)

30-May-2019

Dear Mr Wang:

On behalf of the Editors, I am pleased to inform you that your Manuscript RSOS-181806.R2 entitled "Accurate wisdom of the crowd from unsupervised dimension reduction" has been accepted for publication in Royal Society Open Science subject to minor revision in accordance with the referee suggestions. Please find the referees' comments at the end of this email.

The reviewers and Subject Editor have recommended publication, but also suggest some minor revisions to your manuscript. Therefore, I invite you to respond to the comments and revise your manuscript.

- Ethics statement

- Data accessibility

<http://datadryad.org/submit?journalID=RSOS&manu=RSOS-181806.R2>

- Competing interests

- Authors' contributions

- Acknowledgements

- Funding statement

Because the schedule for publication is very tight, it is a condition of publication that you submit the revised version of your manuscript before 08-Jun-2019. Please note that the revision deadline will expire at 00.00am on this date. If you do not think you will be able to meet this date please let me know immediately.

Kind regards,
Alice Power
Royal Society Open Science

on behalf of Dr Hamed Haddadi (Associate Editor) and Marta Kwiatkowska (Subject Editor)
openscience@royalsociety.org

Subject Editor Comments:

Having read the previous review and the authors' cover letter, I am in agreement with the reviewer that the suggested changes should be implemented. The authors should do their best to address all aspects of the reviewer's feedback and provide a detailed response file outlining how this has been done.

Associate Editor Comments to Author (Dr Hamed Haddadi):

Dear authors,

If you can improve the final version of the article to address the remaining concerns and clarifications of the reviewers, this paper would be suitable for publications. Please try to address as many of these as possible.

Best wishes

Author's Response to Decision Letter for (RSOS-181806.R2)

See Appendix C.

Decision letter (RSOS-181806.R3)

09-Jul-2019

Dear Mr Wang,

I am pleased to inform you that your manuscript entitled "Accurate wisdom of the crowd from unsupervised dimension reduction" is now accepted for publication in Royal Society Open Science.

on behalf of Dr Hamed Haddadi (Associate Editor) and Marta Kwiatkowska (Subject Editor)
openscience@royalsociety.org

Appendix A

Response to reviewers

We would like to thank the reviewers for their valuable comments. Reviewer comments are copied in *blue italic*.

Reviewer 1

Summary

In this paper the authors demonstrate that unsupervised dimensionality reduction techniques can be used to perform wisdom of the crowds style aggregation with performance comparable or superior to that of established techniques. This aggregation can be performed with both categorical and continuous responses, with PCA and Isomap performing well. The paper is generally clear, and the figures illustrative. I have only minor comments I would like to see addressed.

Minor Comments

Pg 5 ln 5 “at least 15 crowd wisdoms of the same dermatologists” - This seems like an odd phrase. I assume this means a pooled population of at least 15.

We apologize for the confusion in English language. We have revised it to “at least 15 crowd wisdoms of those dermatologists”.

Pg 10 ln 45 I would be useful to have a brief definition of AUROC and AUPR on pg 5 when they are first mentioned, pg 10 is a little late.

We have moved the definitions to pg 5.

Figures

In general the ROC and PR curves are hard to distinguish and have lots of white-space. It might be beneficial to include insets for some figures where differences between curves are highlighted in the text, a small alpha on the curves might also be considered to aid the eye in following them. It could also be made clearer what the SML and CUBAN markers in Figure 3 refer to, the text suggests that their performance in each case ends up similar to another technique, but what is being labelled by the stars in the figure is not obvious.

We appreciate the reviewer’s suggestions on figure improvements.

Since the quantitative method comparisons depend mostly on other figures (e.g. Figs 3E and 5B), Figs 3A and 3B aim to provide a more qualitative comparison in a different facet, in combination with the text. Therefore, we tried not to over-interpret or over-visualize the details, especially considering the limited resolution of the dataset of 111 dermoscopy images. We also tried to increase the transparency of the curves, but each curve became less obvious and the overlay of multiple colors also made it hard to distinguish the color combinations.

Reviewer 2

This manuscript describes an unsupervised dimensionality reduction approach to wisdom-of-crowd, meta-classifier learning. The authors propose using the first principal component of the columns of the prediction matrix (classifier-by-data instance) as the consensus prediction. The authors compare their PCA-based approach, along with other standard dimension reduction methods, to supervised techniques as well as two existing methods specialized for this task, SML and CUBAM, on DREAM2 transcription factor activity prediction data and skin cancer classification data. The authors conclude PCA consistently achieves best performance and has the ability to handle continuous confidence scores as input. The manuscript, though succinct, contains extensive computational experiments and is overall clearly written.

Unfortunately, the PCA-based approach proposed by the authors is near identical to the existing work of SML, which puts the novelty of this work under question. Note that SML forms the covariance matrix of the prediction matrix, replaces the diagonal entries, and finds the leading eigenvector. This eigenvector is then multiplied to the prediction matrix to obtain the consensus predictions. If one removes from SML the modification of diagonal elements of covariance matrix, the resulting algorithm is identical to the authors' PCA approach. None of this connection discussed in the manuscript, which seems misleading.

We appreciate the reviewer's insight into the method similarities. We have noted the similarity between SML and PCA after first referring to Fig 5A.

The major contribution and novelty of this paper is linking the problem of crowd wisdom with unsupervised dimension reduction, which is much better studied and known. Crowd wisdom can now be advised by existing researches in unsupervised dimension reduction, such as introducing a class of publicly available crowd wisdom methods. This also includes manifold-based methods, such as Isomap, which may capture potential nonlinear effects beyond SML's capabilities. We have updated the Discussion to reflect the novelty.

Related to this concern, the main motivation that the authors give for using dimension reduction is not sound. The authors assume a probabilistic model of the noisy ensemble of predictions where there is a single latent component that each classifier independently estimates. However, no justification is given as to how particular dimension reduction techniques that the authors consider correctly infer this latent factor, which would depend on the details of the error model and the assumptions of different dimension reduction methods. For example, PCA finds a direction that maximizes variance explained—does this objective coincide with the inference of the latent factor in the authors' model? In this regard, SML appears to be a more principled counterpart to the authors' PCA approach as SML defines a probabilistic model, and relates the leading eigenvector of the modified covariance matrix to the desired latent variable. In fact, the modification of diagonal elements, which may be the only difference between SML and PCA, is theoretically justified.

Given this observation, I was not surprised to see that SML lies exactly on PCA's performance curve in Figure 3. Indeed, the authors note that the performance of SML and PCA converge as more classifiers are added to the data set, which is likely due to the diminishing contribution of the diagonal elements on the covariance matrix on the leading eigenvector.

The class probability, or "latent component" by the reviewer, is the conditional probability of true class assignment given all available information, as defined in text and illustrated in Figs 1 and 2B.

It can be regarded as the theoretically best possible estimator, such as one that achieves Bayes error rate. Therefore, the exact values of the class probability cannot be learned in practice.

However, since the class probability best estimates the true class, and all realistic estimators are effectively estimating the class probability, each estimator's performance with respect to the true class perfectly reflects their performance with respect to the class probability. Therefore, this paper benchmarks the recovery of "latent factor" from Fig 3 through Fig 5, by both unsupervised dimension reduction and existing crowd wisdom methods.

We have clarified the definition of class probability when first referring to Fig 2A.

The authors describe the ability to handle continuous input as a major contribution, but it was not clear to me why the same SML procedure could not be applied to the continuous prediction matrix to be compared with PCA, even if SML is originally designed for binary classification. This comparison should be included in future experiments. In addition, Figure 3 should include the complete ROC or PR curves for SML using the continuous predictions scores before binarization.

This article introduces a class of readily available crowd wisdom methods from unsupervised dimension reduction by revealing their links, rather than by proposing and implementing completely new methods. The targeted audience are ready program users for crowd wisdom problems. A continuous version of SML is not publicly available.

Appendix B

Response to reviewers

We would like to thank the editors for handling our manuscript, and also the reviewer for their comments. We have copied the reviewer comments in *blue italic*, and responded to them below.

Reviewer 2

I agree with the authors that casting crowd-of-wisdom as dimensionality reduction and using off-the-shelf techniques like PCA is a valuable perspective that should see the light of day. However, the way the existing work of SML is presented is misleading for the readers, given that SML is grounded in spectral theory and closely resembles PCA, thus fitting into the same dimensionality reduction framework that the authors describe as novel. In response to this comment the authors have added only a high-level mention of “SML’s similarity with PCA by design,” but this similarity is not explained anywhere in the manuscript.

We appreciate the reviewer’s interest in the similarity between SML and PCA. However, we doubt if such an in-depth discussion would be suitable for this manuscript. Given that none of the title, abstract, or conclusion of this paper mentions SML, such a discussion appears off-topic. In addition, the discrepancy in the numbers of users of PCA and SML suggests that the general readers’ interest is more in applying PCA than in understanding SML. Therefore, we have added an in-place citation of the SML paper (Parisi et al 2013), so readers with the specialized interest can understand the resemblance from the original SML publication.

We also doubt if the potential link between SML and PCA are sufficiently informed and grounded. The original SML paper suggests its roots in probabilistic statistics (e.g. MLE), rather than other known dimension reduction methods. Neither the phrase “dimension reduction” nor “principal component” was even mentioned in the SML paper, suggesting its potential link with PCA or dimension reduction was probably not realized, let alone justified. It is also unclear how SML can reduce the input data to more than one dimension, which is the capability of normal dimension reduction methods.

As a matter of principle, we believe it’s the responsibility of the supporters of a method to extend, implement, and explain the method, rather than those suggesting alternative methods, even if it is beneficial to the research community.

In addition to adding a more detailed discussion of this theoretical connection, the language should be changed throughout the manuscript to place more emphasis on “off-the-shelf” or “standard” dimensionality reduction techniques, which I believe is the main focus of this work. I believe this update will help to emphasize the contribution of the authors.

We would like to thank the reviewer for the suggestion. We have placed more emphasis on “off-the-shelf” and “standard” dimensionality reduction to emphasize the contribution.

In response to my other comment that the point estimates for SML and CUBAM in Figure 3 should be replaced with full curves for more comprehensive comparisons, the authors opted not to make this change and have added an explanation that these methods “only accept and output binary responses.” Although this statement may be pedantically true, all it takes is a cosmetic change to evaluate SML and CUBAM based on their intermediate non-binary scores just before the final binarization of the output, and I have difficulty understanding why the authors are unable to incorporate this change. My

initial review pointed out this possibility for SML only, but I have additionally observed that CUBAM also internally calculates non-binary scores before binarizing them, which can be similarly used to plot the entire PR/ROC curves for more informative comparison with other methods, which I believe is in the interest of the readers.

In addition to all the above, we believe it's unnecessary and practically difficult to fully address the reviewer's comment, also because:

- We've already shown in Fig 5B, that PCA with binary output still outperforms SML and CUBAM. This is sufficient to prove that SML and CUBAM with continuous output are different from PCA.
- Whether SML and CUBAM are asymptotically identical to PCA is not an important question. Even if they were, PCA would still be favored by users due to its simplicity, its availability, and our better understanding of it.

Minor comments: - "Despite some existing applications of principal component analysis (PCA) in specific research contexts ([3, 4, 10]), the general relation between wisdom of the crowd and machine learning was not understood." – This statement seems misleading due to its scope. Instead of "machine learning", I suggest "standard dimensionality reduction techniques."

We appreciate the reviewer's suggestion and have adopted the change.

*- "We found SML to highly correlate with and converge to thresholded PCA on binary/binarized datasets as the number of individuals increased (Figure 5A, Figure S7, Methods)" – Figure S7 states they did *not* converge. The description needs to be updated.*

We apologize for the mis-referencing. The x axis of Fig S7 (current Fig S8) is not the number of individuals. The reference has been removed.

- The last sentence of Results section – "Overall, PCA and Isomap were more reliable and accurate than other dimension reduction methods and previous wisdom of the crowd methods." – does not follow logically from previous text. The same paragraph only describes comparison with supervised/other off-the-shelf methods but not previous wisdom-of-crowd methods. The previous paragraph in fact states that SML/CUBAM were less sensitive (which, as noted, is unsurprising given binarization).

This sentence summarizes the whole paragraph. Fig S10 shows that PCA outperforms SML and CUBAM.

Appendix C

Response to reviewers

We would like to thank the editors for processing our manuscript, and also the reviewer for their comments. We have copied the reviewer comments in *blue italic*, and responded to them below.

Reviewer 2

I agree with the authors that casting crowd-of-wisdom as dimensionality reduction and using off-the-shelf techniques like PCA is a valuable perspective that should see the light of day. However, the way the existing work of SML is presented is misleading for the readers, given that SML is grounded in spectral theory and closely resembles PCA, thus fitting into the same dimensionality reduction framework that the authors describe as novel. In response to this comment the authors have added only a high-level mention of “SML’s similarity with PCA by design,” but this similarity is not explained anywhere in the manuscript.

The potential link between SML and PCA is not sufficiently informed and grounded, and requires extra study. The original SML paper never mentioned the phrase “dimension reduction” or “principal component” and could not realize, let alone demonstrate, the potential link. It is also unclear how SML can reduce the input data to more than one dimension, which is the capability of normal dimension reduction methods.

Claiming this potential link requires extra study, but would not benefit this paper or the whole field. The aim of our manuscript is to introduce dimension reduction methods for crowd wisdom, rather than an in-depth comparison and analysis of different wisdom of the crowd methods. Therefore, understanding SML’s potential link with PCA appears off-topic. Given PCA’s superior performance as shown in this manuscript, and the obvious fact that PCA is much better studied and widely applied than SML, analyzing SML’s methodology alone would not advance the field either.

Since the reviewer could not state the benefit of the study, we have added an in-place citation of the SML paper (Parisi et al 2013) and some extra discussion, so readers with the specialized interest can understand the potential link from the original SML publication.

In addition to adding a more detailed discussion of this theoretical connection, the language should be changed throughout the manuscript to place more emphasis on “off-the-shelf” or “standard” dimensionality reduction techniques, which I believe is the main focus of this work. I believe this update will help to emphasize the contribution of the authors.

We would like to thank the reviewer for the suggestion. We have placed more emphasis on “off-the-shelf” and “standard” dimensionality reduction to emphasize the contribution.

In response to my other comment that the point estimates for SML and CUBAM in Figure 3 should be replaced with full curves for more comprehensive comparisons, the authors opted not to make this change and have added an explanation that these methods “only accept and output binary responses.” Although this statement may be pedantically true, all it takes is a cosmetic change to evaluate SML and CUBAM based on their intermediate non-binary scores just before the final binarization of the output, and I have difficulty understanding why the authors are unable to incorporate this change. My initial review pointed out this possibility for SML only, but I have additionally observed that CUBAM also internally calculates non-binary scores before binarizing them, which can be similarly used to

plot the entire PR/ROC curves for more informative comparison with other methods, which I believe is in the interest of the readers.

This comment — the extension to continuous output in particular — also falls into the “no-benefit” category of study we have responded to above. In addition,

- Fig 5B already shows that PCA with binary output still outperforms SML and CUBAM. This is sufficient to prove that PCA outperforms SML and CUBAM with continuous output, which reinforces the lack of benefit of the study the reviewer requested, and answers the reviewer’s question of whether their performances are the same.
- As mentioned in our previous reply to Reviewer 1 on Fig 3, *“we tried not to over-interpret or over-visualize the details, especially considering the limited resolution of the dataset of 111 dermoscopy images.”*

Minor comments: - “Despite some existing applications of principal component analysis (PCA) in specific research contexts ([3, 4, 10]), the general relation between wisdom of the crowd and machine learning was not understood.” – This statement seems misleading due to its scope. Instead of “machine learning”, I suggest “standard dimensionality reduction techniques.”

We appreciate the reviewer’s suggestion and have adopted the change.

*- “We found SML to highly correlate with and converge to thresholded PCA on binary/binarized datasets as the number of individuals increased (Figure 5A, Figure S7, Methods)” – Figure S7 states they did *not* converge. The description needs to be updated.*

We apologize for the mis-referencing. The x axis of Fig S7 (current Fig S8) is not the number of individuals. The reference has been removed.

- The last sentence of Results section – “Overall, PCA and Isomap were more reliable and accurate than other dimension reduction methods and previous wisdom of the crowd methods.” – does not follow logically from previous text. The same paragraph only describes comparison with supervised/other off-the-shelf methods but not previous wisdom-of-crowd methods. The previous paragraph in fact states that SML/CUBAM were less sensitive (which, as noted, is unsurprising given binarization).

This sentence summarizes the whole paragraph. Fig S10 shows that PCA outperforms SML and CUBAM.